# Multi-Omics and Experimental Validation Identify GPX7 and Glutathione-Associated Oxidative Stress as Potential Biomarkers in Ischemic Stroke

**DOI:** 10.3390/antiox14060665

**Published:** 2025-05-30

**Authors:** Tianzhi Li, Sijie Zhang, Jinshan He, Hongyan Li, Jingsong Kang

**Affiliations:** Key Laboratory of Pathobiology, Department of Pathophysiology, Ministry of Education, College of Basical Medical Sciences, Jilin University, 126 Xinmin Street, Changchun 130021, China; tzli23@mails.jlu.edu.cn (T.L.); sijie23@jlu.edu.cn (S.Z.); hejs24@jlu.edu.cn (J.H.)

**Keywords:** ischemic stroke, oxidative stress, multi-omics, molecular docking

## Abstract

Ischemic stroke (IS) is the leading cause of disability and death worldwide, and its high incidence, disability and recurrence rates impose a heavy economic burden on families and society. Recent studies have shown that oxidative stress plays a key role in the pathophysiological mechanisms of ischemic stroke, not only participating in the onset and development of neuronal damage in the acute phase but also significantly influencing the long-term prognosis of ischemic stroke through molecular mechanisms, such as epigenetic modifications. However, the potential targets of oxidative stress-related genes in IS and their mechanisms of action remain to be elucidated. The aim of this study was to systematically analyse the function and significance of oxidative stress-related genes in IS. We obtained IS-related gene expression datasets from the GEO database and integrated known oxidative stress-related genes from the Genecards database for cross-analysis. Multidimensional feature screening using unsupervised consensus clustering and a series of machine learning algorithms led to the identification of the signature gene *GPX7*. The correlation between this gene and immune cell infiltration was assessed using MCPcounter and a potential therapeutic agent, glutathione, was identified. Binding was verified by molecular docking (MD) analysis. In addition, single-cell RNA sequencing data were analysed to further reveal expression in different cell types and its biological significance. Finally, we performed in vivo experiments using the Wistar rat middle cerebral artery occlusion (MCAO) model, and the results indicated that *GPX7* plays a key role in IS, providing a new theoretical basis and potential intervention target for the precise treatment of IS.

## 1. Introduction

Ischemic stroke (IS) is the second leading cause of death and the first leading cause of disability in adults worldwide, and its epidemiological burden continues to increase [1]. According to the World Health Organization, there are approximately 13.7 million new cases of IS each year worldwide, and the age of onset is significantly younger [2]. Currently, the diagnostic system for IS is mainly based on neuroimaging (e.g., ischaemic penumbra by DWI-MRI) and clinical scales (NIHSS score) [3,4]. Although reperfusion therapies, such as intravenous thrombolysis (IV-tPA) [5] and endovascular thrombolysis (EVT), have significantly improved the prognosis in the acute phase [6], the overall disability rate of IS is still more than 68%, mainly limited by a narrow window of opportunity for treatment (<4.5 h) and an absence of reperfusion of up to 23% [7]. This clinical dilemma stems from the complexity of the pathogenesis of IS; in addition to acute glutamate excitotoxicity, calcium overload and mitochondrial dysfunction, oxidative stress and inflammation are central factors in secondary brain injury [8]. Given this complexity of factors and their interactions, the selection of therapeutic development strategies targeting oxidative stress is becoming increasingly important.

During ischaemia-reperfusion injury, the overproduction of reactive oxygen species (ROS) is a central driver of neuronal damage. First, ROS induce lipid peroxidation of cellular phospholipids, which produces toxic metabolites, such as malondialdehyde (MDA), and disrupts membrane structure [9]. Secondly, ROS interfere with cell signalling and energy metabolism through oxidative modification of key functional proteins (e.g., protein kinase C, mitogen-mitogenin-mitogenin) [10]. In addition, damage to the mitochondrial electron transport chain further exacerbates ROS accumulation, resulting in a vicious cycle of ROS-induced ROS that ultimately leads to ATP depletion and programmed neuronal death [11]. Therefore, in-depth analysis of the molecular regulatory network of oxidative stress and identification of key intervention targets are important for accurate diagnosis and treatment of IS.

To explore the role of oxidative stress in ischemic stroke (IS) in depth, this study combined multi-omics data analysis and experimental validation to conduct a systematic investigation. First, oxidative stress-related genes (OXs, threshold score > 7) were screened based on the GeneCards database, and differential expression analysis and weighted gene co-expression network analysis (WGCNA) were performed in combination with transcriptome data from IS patients in the Gene Expression Omnibus (GEO) database to identify the core oxidative stress modules associated with IS. Subsequently, patients were classified into two molecular subtypes by consensus clustering (CC) and their correlation with immune cell infiltration was analysed using the MCPcounter algorithm. Based on this, three machine learning algorithms, including Lasso Regression, Support Vector Machine Recursive Feature Elimination (SVM-RFE) and Random Forest, were used to further screen for the key oxidative stress regulator *GPX7* and validate its expression specificity in an independent external dataset. Glutathione, a potential regulatory drug targeting *GPX7*, was retrieved from Drugbank and the robust binding capability of both was assessed by molecular docking analysis. To assess the functional relevance of GPX7 in vivo, its transcript levels were further analysed using a rat middle cerebral artery occlusion (MCAO) model combined with qRT-PCR. The aim of this study was to elucidate the role of the oxidative stress-related gene GPX7 as a potential therapeutic target in ischemic stroke and to assess the therapeutic potential of glutathione as a novel intervention strategy.

## 2. Materials and Methods

### 2.1. Data Acquisition and Processing

In this study, peripheral blood RNA sequencing data of IS patients were obtained from the GEO database (https://www.ncbi.nlm.nih.gov/geo/ accessed on 20 March 2025.) for systematic analysis of the function of oxidative stress-related genes and their role in ischemic stroke (IS). Among them, GSE16561 (GPL6883 platform) was selected as the master dataset for further analysis, while GSE58294 (GPL570 platform) was used as a test dataset to verify the reliability of the results. In addition, the single-cell RNA sequencing dataset GSE174574 was selected for subsequent validation analyses to assess the expression profiles of key genes and their biological significance in different cell types. To integrate known oxidative stress-related genes (OXs), 1719 oxidative stress-related genes were retrieved from the GeneCards database (https://www.genecards.org/ accessed on 23 March 2025) [12], and a screening threshold with a score greater than 7 was set to ensure that the selected genes had high relevance and confidence. To further quantify the enrichment of oxidative stress-related genes in the samples, we calculated gene enrichment scores (IS scores) on the test set (GSE58294) using the ssGSEA algorithm in the R package gsva and assessed their correlation with changes in clinical function. The flow of the study is shown in Figure 1.

### 2.2. Identification and Functional Annotation of Differentially Expressed Genes (DEGs)

To identify differentially expressed genes (DEGs) between IS patients and healthy controls, the GSE16561 dataset was normalised and analysed for differences using the ‘limma’ R software package (version 3.56.2) [13]. The screening criteria were set at adjusted *p*-value < 0.05 and |log2FC| > 0.5 to ensure that the screened genes had significant expression differences and biological significance. Subsequently, DEGs were visualised using R packages, such as pheatmap (version 1.0.12), dplyr (version 2.4.0), ggplot2 (version 3.5.0) and ggrepel (version 0.9.4), including heatmaps, volcano plots and cluster analyses, to visually present the gene expression patterns and differential distributions.

### 2.3. Weighted Gene Co-Expression Network Analysis (WGCNA)

In order to identify gene modules highly correlated with IS, we performed weighted gene co-expression network analysis (WGCNA) on the GSE16561 dataset using the R package ‘WGCNA’ (version 1.72-5) [14]. First, the top 10,000 genes with the highest variance were selected, and the Pearson correlation coefficients between the samples were calculated to construct a neighbour-joining matrix. The neighbour-joining matrix was transformed into a topological overlap matrix (TOM) by a soft-threshold power parameter β = 8 to capture the co-expression relationships among genes. Subsequently, gene modules with similar expression patterns were identified using a dynamic tree-cutting approach, and the modular genes most significantly associated with IS were filtered out by calculating the correlation between modular genes (MEs) and phenotypes (IS states). These modular genes were further cross-analysed with DEGs and OXs in the GeneCards database, and the hub genes significantly associated with IS were finally identified.

### 2.4. Functional Enrichment Analysis

In order to reveal the unique functional characteristics of hub genes and their potential biological significance, we performed Gene Ontology (GO) enrichment analysis and Kyoto Encyclopedia of the Genome (KEGG) pathway analysis of hub genes using the ‘clusterProfiler’ R software package (version 4.8.3) [15].The GO analysis covered the following domains: biological process (BP), molecular function (MF) and cellular component (CC). KEGG analysis focused on the regulatory mechanisms of signalling pathways. The screening criterion was an adjusted *p*-value of less than 0.05, which was considered to have a significant enrichment level. The enrichment analyses provided a comprehensive understanding of the potential role of hub genes in the pathophysiological mechanisms of IS and the key biological processes in which they are involved.

### 2.5. Consensus Clustering Analysis and Subtype Differences

Based on the expression profiles of the hub genes, we performed unsupervised consensus clustering analysis in the test set (GSE58294) to identify different molecular subtypes of IS. By GSVA enrichment analysis, we compared the differences in gene enrichment scores among subtypes and statistically analysed the results using t-tests. In addition, in order to comprehensively analyse the IS-associated immune microenvironment, we performed immune infiltration analyses on the test set using the ‘MCPcounter’ R package (version 1.2.0) [16] and assessed the infiltration levels of various immune cell types in different subtypes. These analyses not only revealed the molecular characteristics of the different subtypes but also provided insights into the potential role of immune cell populations in the pathophysiology of IS.

### 2.6. Machine Learning Algorithms for GPX7 Identification

In order to select the most representative feature genes from the hub genes, three classical machine learning algorithms for multidimensional feature selection were used in this study. Firstly, the Least Absolute Shrinkage and Selection Operator (LASSO) algorithm was used to downscale the hub genes, and model construction and optimisation was achieved using the ‘glmnet’ R package (version 4.1-2) [17]. The LASSO regression determines the optimal regularisation parameter λ through cross-validation methods to maximise the predictive performance of the model and retain key variables. Secondly, a Support Vector Machine Recursive Feature Elimination (SVM-RFE) model was used to further optimise the selection of feature genes. We used the ‘e1071’ R package (version 1.7-13) to construct the SVM-RFE model [18] and evaluated the average misclassification rate through 5-fold cross-validation to gradually eliminate redundant features and retain the most discriminative genes. Finally, based on the random forest (RF) algorithm, the RF model was constructed using the ‘randomForest’ R package (version 4.6-14) [19], and the predictive ability of each gene was evaluated by calculating the mean decrease accuracy (MDA). Finally, the overlapping biomarker *GPX7* identified by the three algorithms was designated as the signature gene of oxidative stress-related genes in IS, indicating its central position in the pathological mechanism of IS.

### 2.7. Comparative Analysis of GPX7 High and Low Expression Groups

To further explore the functional significance of *GPX7*, we categorised the samples into GPX7 high and low expression groups based on the GSE58294 test set and analysed the samples using the ‘limma’ R package (version 3.56.2) and the ‘ggplot2’ R package (version 3.5.0). An R package (version 3.5.2) was used to analyse the samples. An R package (version 3.56.2) and the ‘ggplot2’ R package (version 3.5.0) were used to analyse and display the differentially expressed genes (DEGs) between the two groups. Screening thresholds were set at |log2FC| > 0.5 and *p* < 0.05. Gene Ontology (GO) enrichment analyses were then performed on the up- and down-regulated genes and the ‘clusterProfiler’ R package (version 4.8.3) was used to reveal the biological processes and molecular functions in which these genes may be involved.

At same time, to validate the disease prediction ability of GPX7, we plotted receiver operating characteristic curves (ROCs) in our test set and measured key performance metrics, including area under the curve (AUC), sensitivity and specificity. The results indicate that GPX7 has high diagnostic efficacy in distinguishing IS patients from healthy controls. In addition, by integrating the miRTarBase database through the NetworkAnalyst platform (https://www.networkanalyst.ca/ accessed on 30 March 2025.) [20], we predicted the potential regulatory miRNAs of *GPX7* and visualised the regulatory network using Cytoscape software (version 3.9.1) [21].

### 2.8. Single-Cell Data Analysis

In order to verify the expression characteristics of *GPX7* in different cell types and its biological significance, we performed systematic analysis of the single-cell RNA sequencing dataset GSE174574. Specific quality control procedures were as follows: genes with less than 300 or more than 5000 counts and genes expressed in less than 3 cells were excluded; genes with more than 10% mitochondria-derived unique molecular identifier (UMI) counts were filtered out. After log-normalisation of the dataset, the top 3000 high variant genes (HVGs) were selected for correlation analysis. The data were then normalised using the ScaleData function and subjected to principal component analysis (PCA). Cells were clustered according to the top 20 principal components with a resolution parameter of 0.5, and t-SNE and UMAP plots were generated to visualise cell distribution. To annotate cell types, we performed automatic annotation of cell clusters using the GPT4type tool [22]. In addition, in order to infer and analyse intercellular communication and better understand intercellular signalling and regulatory mechanisms, we analysed the intercellular interaction network using the CellChat software package (v1.6.1) [23]. The results show that *GPX7* exhibits significant differences in exocytosis in fibroblasts, suggesting its potential regulatory role in the IS cellular environment.

### 2.9. Drug Prediction and Molecular Docking

To explore potential therapeutic compounds targeting *GPX7*, we first uploaded key genes into the Drugbank database and screened glutathione (GSH) as a potential active therapeutic molecule. Then, we obtained the 3D structure of GPX7 from the UniProt database (https://www.uniprot.org/ accessed on 3 April 2025.) and pre-processed the protein to optimise its suitability for subsequent analysis. Meanwhile, the structural data file (SDF format) of glutathione was downloaded from PubChem database (https://pubchem.ncbi.nlm.nih.gov/ accessed on 4 April 2025.) and converted to mol2 format using Open Babel software (3.1.1)The docking analysis of GPX7 and glutathione was then performed using AutoDock molecular docking software (v4.2.6) to predict their binding modes and binding free energies, and the most stable docked conformations were visualised using PyMOL (3.0).To further assess the stability of binding, molecular dynamics simulations were performed using GROMACS (2023.2) [24] to analyse the dynamic behaviour and interaction characteristics of the complexes under physiological conditions.

### 2.10. pMCAO Model Construction

Since ischemic stroke occurs mainly in the elderly population, this study used the permanent middle cerebral artery occlusion (pMCAO) model to simulate ischemic stroke in aged rats. Sixteen male Wistar rats (26–28 months old) were obtained from Beijing Viton Lihua Laboratory Animal Co. The rats were randomly divided into two groups: the sham-operated group (8 rats) and the pMCAO group (8 rats). The specific pMCAO procedure was as follows: anaesthesia was induced with 5% isoflurane and maintained with 2% isoflurane by inhalation. A midline neck incision was made to expose the right common carotid artery (CCA), external carotid artery (ECA) and internal carotid artery (ICA). In the pMCAO group, the ECA was ligated, and a nylon wire was inserted through the CCA into the ICA to occlude the origin of the middle cerebral artery (MCA). The sham-operated group underwent the same procedure but without insertion of the nylon wire. Ischaemia was maintained for 24 h to simulate permanent cerebral infarction without reperfusion. Rats were then euthanised by intraperitoneal injection of sodium pentobarbital (150 mg/kg) followed by saline infusion via cardiac catheter. Plasma was rapidly collected for RNA extraction as well as for determination of GPX7 transcript levels. All experimental protocols were approved by the Animal Care and Use Committee of the School of Basic Medical Sciences, Jilin University, and the Guidelines for the Care and Use of Laboratory Animals were strictly adhered to.

### 2.11. qRT-PCR Analysis

Total RNA was extracted from rat peripheral blood using the TRIzol method, and first-strand cDNA was synthesised using the First Strand Synthesis Master Mix (LABLEAD, Beijing, China). Primers used for cDNA amplification are listed in Appendix A. cDNA was then mixed with SYBR Premix Ex Taq2 (TaKaRa, Japan) and synthesised primers and subjected to real-time quantitative PCR. PCR conditions were selected according to the manufacturer’s protocol as follows: 2 min at 50 °C; 10 min at 95 °C; 45 cycles of 10 s at 95 °C, 10 s at 60 °C and 15 s at 72 °C. PCR conditions were selected according to the manufacturer’s protocol. Finally, mRNA expression levels were normalised to the expression level of the internal reference GAPDH.

### 2.12. Statistical Analysis

Statistical analyses were performed using R software version 4.3.0. Normality of the data was assessed before analyses. Comparisons between the two experimental groups (disease and control) were performed using a two-tailed unpaired Student’s *t*-test. Pearson’s correlation analysis was performed to assess the association between the expression levels of key genes and immune cell infiltration. *p* values less than 0.05 were considered statistically significant.

## 3. Results

### 3.1. IS Can Be Stratified by Oxidative Stress-Related Genes

To develop a gene signature-based stratification method for ischemic stroke (IS), we extracted 1719 oxidative stress-related genes from the GeneCards database and calculated the expression profiles of these genes in the GSE58294 dataset by scoring them using the ssGSEA (single-sample gene set enrichment analysis) method. As shown in Figure 2A, ssGSEA analysis revealed the expression patterns of these oxidative stress-related genes. Further statistical analyses showed that the oxidative stress scores in the patient group were significantly higher than those in the control group (*p* = 0.047; Figure 2B). This result suggests that oxidative stress-related scores have the potential to discriminate between IS patients and healthy individuals and can be used for molecular stratification of IS.

### 3.2. Differentially Expressed Genes (DEGs) in the GSE16561 Cohort

We performed differential expression analysis on the GSE16561 dataset to identify differentially expressed genes (DEGs) between IS patients and healthy controls. The results showed that a total of 581 genes exhibited significant differential expression, including 279 up-regulated genes and 302 down-regulated genes (Figure 3A,B). The expression patterns of these DEGs were visualised by heatmaps and volcano plots, clearly demonstrating the differences in gene expression between IS patients and healthy individuals.

### 3.3. WGCNA Analysis to Identify Hub Genes

To further screen the hub genes most relevant to IS, we applied weighted gene co-expression network analysis (WGCNA) to the GSE16561 dataset. First, using a soft threshold selection (β = 8), we constructed a gene co-expression network close to a scale-free distribution (Figure 4A). Then, using the topological overlap matrix (TOM), we classified the genes into 15 modules and evaluated the correlation of each module with clinical features (Figure 4B–D). Among them, the green-yellow module (correlation coefficient = 0.61, *p*-value = 1 × 10^−200^) was found to have the strongest correlation with IS (Figure 4E). Screening of genes in this module resulted in 3183 significant genes (correlation coefficient > 0.5). Combined with the intersection analysis of DEGs and oxidative stress-related genes, we further narrowed down and identified 20 candidate hub genes (Figure 4F).

### 3.4. Enrichment Analysis Reveals the Functions of Hub Genes

To explore the role of hub genes in the pathophysiological mechanisms of IS, we performed GO (Gene Ontology) and KEGG (Kyoto Encyclopedia of Genes and Genomes) functional enrichment analyses. The results of the GO analyses indicated that, in terms of biological processes (BPs), hub genes were significantly involved in the cellular response to oxidative stress, positive regulation of T-cell receptor signalling pathways, mitochondrial ATP synthesis-coupled electron transport and mitochondrial electron transport chain-associated responses. In terms of biological processes (BPs), hub genes were significantly involved in the cellular response to oxidative stress, positive regulation of T cell receptor signalling pathways, mitochondrial ATP synthesis-coupled electron transport and mitochondrial electron transport chain-related reactions. In terms of cellular components (CCs), hub genes were mainly localised in subcellular structures, such as mitochondrial respiratory bodies, respiratory chain complexes (e.g., mitochondrial respiratory chain complex I) and mitochondrial inner membrane protein complexes. In terms of molecular function (MF), the hub genes exhibited antioxidant activity, NADH dehydrogenase activity and tumour necrosis factor receptor binding (Figure 5A–C), and KEGG pathway analysis further revealed that the hub genes were significantly enriched in the primary immunodeficiency, oxidative phosphorylation and T cell receptor signalling pathways (Figure 5D). These results suggest that hub genes play an important role in the pathophysiological mechanisms of IS by regulating key biological processes, such as oxidative stress response, mitochondrial energy metabolism and immune signalling.

### 3.5. Subtype Classification and Immune Infiltration Analysis

To verify whether the hub genes were effective in molecular stratification of IS, we performed unsupervised cluster analysis of the expression profiles of the hub genes using the ConsensusClusterPlus R software package. The results showed that the hub gene could classify IS patients into two distinct subtypes (cluster 1 and cluster 2; Figure 6A–C). Further analysis showed that cluster 2 had a significantly higher IS score than cluster 1 (*p* = 2.36 × 10^−10^; Figure 6D). This suggests a significant difference in the hub gene expression pattern between the two clusters.

Given the important role of the immune response in the pathogenesis of IS, we analysed the infiltration levels of 10 immune cell types in both subtypes using the MCPcounter algorithm. The results showed that these hub genes showed strong correlations with several immune cell types (e.g., T cells, macrophages, etc.) (Figure 6E,F). In addition, we specifically analysed the correlation between *GPX7* and immune cells and found that it was closely associated with multiple immune cell types (Figure 6G).

### 3.6. Feature Gene Selection

To further identify the most relevant feature genes for oxidative stress in IS, we applied the above 20 candidate genes to three commonly used feature selection algorithms: SVM-RFE (Support Vector Machine Recursive Feature Elimination), RF (Random Forest) and LASSO (Least Absolute Shrinkage and Selection Operator). Using LASSO regression analysis, we selected *GPX7* as the most representative feature gene (Figure 7A). The Random Forest method ranked the importance scores of the 20 candidate genes and selected the top 10 feature genes for further analysis (Figure 7B). 18 genes were tagged by SVM and achieved a prediction accuracy of 0.741 (Figure 7C). Finally, by intersection analysis of the results of the three methods, we confirmed that *GPX7* is a core feature gene of oxidative stress in IS (Figure 7D).

### 3.7. Functional Enrichment Analysis of GPX7-Related Genes and Their Predictive Ability

To further explore the biological functions and potential applications of *GPX7*, we screened the protein-coding genes significantly related to *GPX7* and performed a systematic functional enrichment analysis. The results showed that a total of 310 genes were differentially expressed between the *GPX7* high- and low-expression groups, of which 111 genes were up-regulated (35.8%) and 199 genes were down-regulated (64.2%) (Figure 8A,B). The negatively associated genes were mainly involved in T cell differentiation and regulation of immune response, whereas the positively associated genes were significantly enriched in processes related to cellular immune response and transmembrane transport (Figure 8C,D). In the validation cohort, the prediction model based on *GPX7* showed good discriminatory ability (AUC = 0.649, 95% CI: 0.529–0.762), suggesting its potential as an IS biomarker (Figure 8E). In addition, *GPX7* expression levels were significantly correlated with several immune cell types, such as endothelial cells, T cells and monocytes (Figure 6G). Predicted by the NetworkAnalyst database, we found several conserved miRNA binding sites in the 3′UTR region of *GPX7*, including miR-205-5p, miR-182-5p, miR-155-5p and miR-34a-5p, which have been shown to be involved in the pathophysiological process of IS (Figure 8F). This finding not only validates the reliability of the *GPX7* regulatory network but also provides a new perspective for a deeper understanding of the mechanism of oxidative stress in IS.

### 3.8. Single-Cell Transcriptome Data Analysis Reveals Cellular Heterogeneity

After extracting single-cell transcriptome data from the GSE174574 dataset, we performed rigorous quality control (QC) screening of the samples. Specifically, a threshold range was established based on the following key metrics: 300 < nFeature_RNA < 5000 and percent.mt < 10 to exclude low quality cells (e.g., lysed or apoptotic cells) as well as aberrant cells due to technical errors (Figure 9A–C). The standard deviation of gene expression was then calculated and the top 10 genes with the highest variability were displayed; these highly variable genes tend to reflect regulators of key biological processes (Figure 9D). After QC, the data were normalised and log-normalised to remove technical noise. Next, highly variable genes were downscaled by principal component analysis (PCA) and unsupervised clustering analysis was performed based on the top 20 principal components (PCs). To optimise the clustering results, the resolution parameter was set to 0.4, resulting in a fine delineation of cellular subpopulations. In addition, to eliminate the potential experimental batch effect, the Harmony algorithm was used to batch correct and integrate the data to ensure consistency of biosignals between different samples (Figure 9E,F).

To further reveal the heterogeneity between cells, we projected the data into UMAP space for non-linear downscaling and visualisation analysis. Through this process, a total of 25 different cell clusters were identified and functionally annotated using the Gpt4celltype toolkit and classified into 17 major cell types, including antigen-presenting cells, astrocytes, biliary epithelial cells, cerebrovascular cells, endothelial cells, fibroblasts, macrophages and glia, endothelial cells, fibroblasts, macrophages, microglia, monocytes, neural progenitor cells, neutrophils, oligodendrocytes, granulocytes and granulocytes, oligodendrocytes, pericytes, photoreceptor cells, smooth muscle cells, T cells and vascular smooth muscle cells (Figure 10A–C). In addition, histograms of the cellular proportions of the classical markers were plotted to visualise the composition of different cell types (Figure 10D), which further verified the reliability of the annotation results. The UMAP visualisation results showed that the expression of GPX7 was mainly enriched in fibroblasts (Figure 10E,F), suggesting that *GPX7* may play an important role in fibroblast-associated biological processes.

A population of cells with high *GPX7* expression was specifically identified and isolated into a separate subpopulation for further study. The subpopulation was further analysed by detailed annotation (Figure 11A,B) and screened for highly variable genes (HVGs) to reveal their potential functional specificity and biological significance (Figure 11C). On this basis, cell-cell communication (CCC) analysis was performed in conjunction with the subpopulation annotation results to investigate the interaction network between the *GPX7* highly expressed subpopulation and other cell types. The results showed that this subpopulation formed complex interactions with other cell types through multiple ligand–receptor pairs, suggesting that it may play an important role in intercellular signalling (Figure 11D–F). These findings provide important insights into the function of *GPX7* in specific cell subpopulations and its regulatory mechanisms in the tissue microenvironment.

### 3.9. Molecular Docking and In Vivo Validation

To explore the potential of GPX7 as a therapeutic target, this study successfully identified glutathione (GSH) as a potential binding compound through a combination of computational and experimental approaches. Using molecular docking technology (AutoDock Vina), we predicted the binding mode of glutathione and GPX7 protein complex and calculated its binding free energy (ΔG = −7.8 kcal/mol), indicating that the two have significant molecular recognition ability. To further elucidate their molecular mechanisms of action, the optimal docked conformations were analysed using PyMOL 2.4.0 for 3D visualisation, revealing the precise positioning of glutathione in the GPX7 catalytic pocket and its interactions with key amino acid residues (Figure 12A). The stability of the complex was then assessed by molecular dynamics (MD) simulations, and RMSD analysis showed that the protein-ligand complex reached dynamic equilibrium after 10 ns with a low range of fluctuation, suggesting that the binding of the two had good stability. Root Mean Square Fluctuation (RMSF) analysis further revealed the changes in flexibility of each amino acid residue in the protein, which provided an important basis for identifying the flexible and rigid regions in the molecule. Hydrogen bonding, as one of the key forces in protein-ligand binding, reflects the strength of electrostatic interactions. As can be seen in the figure, the number of hydrogen bonds between glutathione and GPX7 basically remains in the range of 1–5 during the simulation, reflecting the stable non-covalent interaction. The radius of gyration (Rg) is used to measure the compactness of the overall structure of the molecule, and its smaller value indicates a more compact molecular structure. The Rg of the complexes remained stable throughout the simulation, indicating that no significant loosening or dissociation of the overall structure occurred. In addition, Buried Solvent Accessible Surface Area (Buried SASA) analysis showed that the contact area of glutathione and GPX7 remained essentially constant throughout the simulation, further verifying the stability of the binding between the two (Figure 12B–G).

In order to further verify the potential role of *GPX7* in pathophysiology, we examined the expression levels of *GPX7* mRNA in the lesion group and the sham-operated group using qRT-PCR (SYBR Green method), and we also examined the expression of *IL6* and *IL1-beta* to demonstrate the success of the cerebral ischemia model construction. The results showed that the expression of IL1-beta and IL6 was elevated, demonstrating the success of the cerebral ischaemia model construction (Figure 12I,J), while the expression of GPX7 mRNA was significantly upregulated (Figure 12H). In conclusion, this study systematically revealed the binding mechanism of glutathione and *GPX7* and its potential function in the disease through molecular docking, molecular dynamics simulation and experimental validation, which provides a solid theoretical basis for the development of subsequent targeted drugs.

## 4. Discussion

Ischemic stroke accounts for approximately 87% of all strokes and is the second leading cause of death worldwide after ischemic heart disease [25]. According to the World Health Organization (WHO), it affects approximately 15 million people worldwide each year, causing more than 6 million deaths and a further 5 million people with permanent disability [26]. Its incidence is increasing with the ageing of the population and the prevalence of metabolic diseases (e.g., hypertension, diabetes mellitus). Although early recanalisation can save ischaemic semidomedullary neurons, the oxidative stress and inflammatory response induced by reperfusion itself has become a point of therapeutic paradox, driving research into neuroprotective agents and antioxidant therapies. Oxidative stress is not only the initiator of ischaemia-reperfusion injury but also a key amplifier throughout the pathology, and its importance is reflected in the following aspects. Firstly, the blockage of the electron transport chain during ischaemia leads to an increase in ROS production, and the superoxide dismutase (SOD) and glutathione (GSH) systems are infiltrated by ROS, which attack the phospholipids (e.g., phosphatidylcholine) of the cell membrane, triggering a chain reaction of lipid peroxidation and destroying the integrity of the cell membrane) [27], and attract neutrophil infiltration by activating key signalling pathways, such as nuclear factor-κB (NF-κB) and Nrf2/ARE. These inflammatory mediators further generate reactive oxygen species (ROS), creating a vicious cycle of ‘inflammation-oxidative stress’ that exacerbates microvascular injury and blood-brain barrier (BBB) disruption [28]. In addition, oxidative stress not only leads directly to neuronal apoptosis (a 3- to 5-fold increase in caspase-3 activation) but also promotes astrocyte proliferation and glial scar formation by preventing neuronal regeneration [29]. In addition, ROS-mediated cerebral oedema and calcium overload can further expand the infarct size and cause irreversible neurological deficits [30]. However, the failure of conventional antioxidants in clinical trials suggests that scavenging ROS alone may not be sufficient to reverse complex pathological networks.

Based on the above research background, we combined bioinformatics analysis and experimental validation to systematically investigate the pivotal role of oxidative stress-related genes in ischemic stroke (IS). First, we obtained peripheral blood transcriptome data of IS patients from the GEO database and integrated 1719 oxidative stress-related genes with score ≥7 from the GeneCards database for feature differentiation and intersection analysis. KEGG enrichment analysis showed that these genes were significantly enriched in oxidative stress and immune-related pathways, suggesting their key roles in the pathological process of IS. Second, through hub gene network construction and unsupervised consensus clustering (k = 2), we classified IS patients into two subtypes and found significant differences in oxidative stress levels and immune microenvironment characteristics between the subtypes. *GPX7* was further investigated as a signature gene using various machine learning algorithms, including LASSO regression, random forest and support vector machine recursive feature elimination (SVM-RFE). The analyses showed that the expression level of *GPX7* was significantly correlated with a variety of immune cells (e.g., T cells, neutrophils and endothelial cells). The results of the independent validation set confirmed that *GPX7* has a high predictive ability in the diagnosis of IS.

To explore the regulatory mechanism of *GPX7* in depth, we performed miRNA regulatory network analysis and identified 47 miRNAs involved in the post-transcriptional regulation of *GPX7*. Among them, miR-205-5p, miR-182-5p, miR-155-5p and miR-34a-5p have been widely reported to play important roles in the pathological processes of IS. For example, in a cerebral ischemia-reperfusion model, miR-155-5p expression was upregulated, leading to accumulation of oxidative stress and blockade of autophagy through inhibition of Nrf2 (a key factor in antioxidant response) and ATG12 (an autophagy-related gene), which exacerbated neuronal injury and infarct expansion [31]. In addition, miR-34a-5p increases p53 acetylation levels by targeting SIRT1 (deacetylase), which in turn upregulates the pro-apoptotic genes PUMA and BAX, leading to neuronal death [32]. These results further validate *GPX7* as a potential therapeutic target. Single-cell data analysis revealed the expression of Gpx7 in different cells. We then retrieved GPX7’s potential target, glutathione, from the Drugbank database and performed molecular docking analysis. The results showed that the two had a low binding energy, indicating a good binding ability and providing new insights into the development of novel antioxidants. It provides an important theoretical basis for the early diagnosis of IS and the development of targeted intervention strategies.

GPX7 (glutathione peroxidase 7) is an important member of the glutathione peroxidase family whose main function is to protect cells from oxidative damage by catalysing the reduction of hydrogen peroxide (H₂O₂) and organic peroxides by glutathione (GSH) [33]. Unlike other members of the GPX family, GPX7 is located in the endoplasmic reticulum (ER) and acts as an oxidative stress sensor and signalling molecule to maintain redox homeostasis through the following mechanisms: First, GPX7 is able to reduce intracellular levels of reactive oxygen species (ROS) by directly neutralising H₂O₂ (independently of GSH) [34]; Secondly, GPX7 coordinates the cellular stress response by interacting with proteins, such as GRP78 and CPEB2 to transmit oxidative stress signals to the unfolded protein response (UPR) and RNA metabolism pathways [35]. In cerebral ischaemia, GPX7 may play an important role by scavenging ROS, inhibiting iron death and synergising with the Nrf2 pathway. In contrast, Nrf2 is able to transcriptionally up-regulate a variety of antioxidant enzymes, including members of the glutathione peroxidase (GPX) family [36]. Although GPX7 has not been clearly demonstrated to be a direct target gene of Nrf2, its localisation to the endoplasmic reticulum and redox-regulatory function suggests that there may be a functional synergy between the two [37]. In our experimental model, the protective effect exhibited by GPX7 may partly involve the activation of the Nrf2 pathway, which may be directly regulated by GPX7 or through a feedback mechanism triggered by the reduction of ROS levels. It has been shown to support the hypothesis that alleviation of oxidative stress stabilises Nrf2 and enhances its nuclear translocation capacity, thereby strengthening antioxidant defence [38,39]. Future studies should further resolve the precise molecular interactions between GPX7 and Nrf2, which may provide new insights into therapeutic strategies for ischaemic stroke based on redox regulation.

Glutathione (GSH) is one of the major non-enzymatic antioxidants in the cell, consisting of glutamate, cysteine and glycine, and exists in both reduced (GSH) and oxidised (GSSG) forms [40]. GSH protects cells from oxidative damage by directly scavenging ROS (e.g., H₂O₂, -OH and lipid peroxides) and supports the function of other antioxidant systems (e.g., thioredoxins) by maintaining a high GSH/GSSG ratio (about 100:1 in normal cells) through glutathione reductase (GR) regeneration [41]. In cerebral ischemia, the synergistic action of GSH and GPX7 exerts a protective effect through the following mechanisms: GPX7 catalyses the reduction of peroxides, thereby preventing ROS accumulation-induced ERS and UPR [42]; at the same time, GPX7 inhibits ERS-associated apoptotic signals (e.g., CHOP, caspase-12) via a GSH-dependent pathway [43], thereby protecting neuronal survival. The molecular docking results showed that the two have good binding ability, and thus, GSH may play a role as a novel antioxidant in the pathological mechanism of cerebral ischemia.

In conclusion, the present study systematically revealed for the first time the closely related regulator of oxidative stress in ischemic stroke (IS), *GPX7*, at a multi-omics level. Through the integration of bioinformatics analysis, machine learning algorithms and experimental validation, we not only clarified the central role of *GPX7* in oxidative stress regulation but also found that it plays a role in neuronal protection, endoplasmic reticulum stress relief and iron death inhibition. stress relief and potential function in iron death inhibition. In addition, we further validated the potential of glutathione (GSH) as a *GPX7*-targeting drug based on the screening and molecular docking analyses of the Drugbank database and confirmed the high-affinity binding ability between the two at the data level. Future studies should focus on the in-depth elucidation of the synergistic mechanism of *GPX7* with other antioxidant enzymes (e.g., GPX4, SOD) and signalling pathways, as well as the design of small molecule activators or gene therapy vectors targeting *GPX7* and the evaluation of their efficacy in animal models and clinical trials.

Although this study reveals the potential role of GPX7 and its associated oxidative stress pathway in ischemic stroke, certain limitations remain. Firstly, the dataset used lacked detailed clinical information, which may have a confounding effect on the ssGSEA score and gene expression analysis. Second, the validation of GPX7 was mainly based on qRT-PCR and molecular docking analyses, which need to be further supported by protein level and functional intervention experiments. Future studies should combine cellular and animal model experiments with clinically annotated complete datasets to systematically explore the mechanism and promote translational applications.

## 5. Conclusions

In this study, we preliminarily revealed a potential association between GPX7 and oxidative stress status in ischemic stroke (IS) model by integrating multi-omics analysis and experimental validation. In vivo results showed that GPX7 expression was significantly changed in the IS animal model, suggesting that it may be involved in the regulation of oxidative stress. These findings provide a new direction for subsequent in-depth studies on the function of GPX7 in cerebral ischemic injury. In addition, a certain regulatory effect of GSH intervention on GPX7 expression was observed in the study, suggesting its potential in antioxidant therapy, which warrants further evaluation of its therapeutic value in subsequent studies.

## Figures and Tables

**Figure 1 antioxidants-14-00665-f001:**
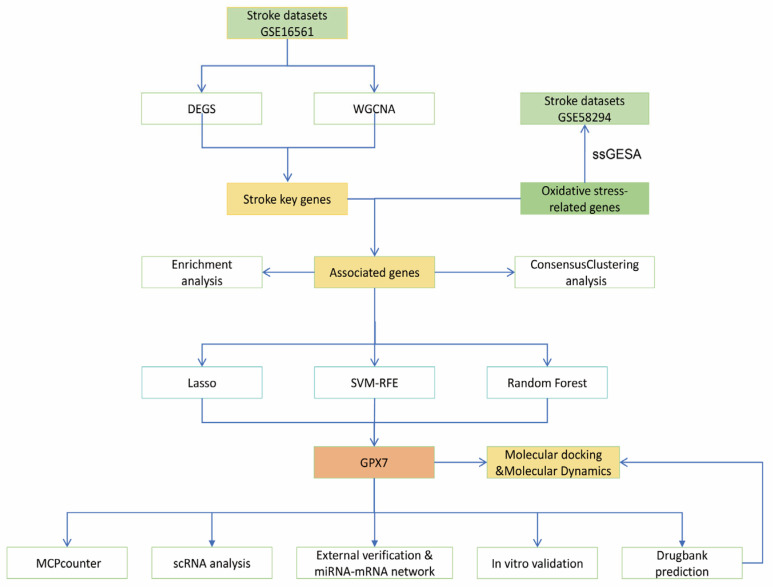
Study flow.

**Figure 2 antioxidants-14-00665-f002:**
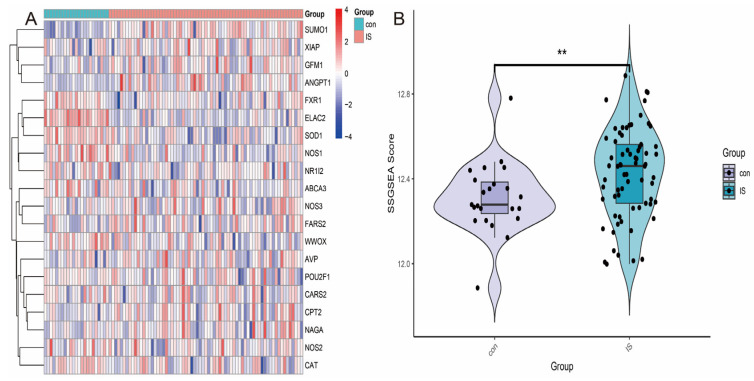
Differences in the expression of oxidative stress-related genes between ischemic stroke (IS) and control groups and score analysis. (**A**) Heatmap of the expression profiles of oxidative stress-related genes in the GSE58294 dataset. (**B**) The oxidative stress score in the IS patient group was significantly higher than in the control group (*p* = 0.047). ‘**’ represents *p* < 0.01.

**Figure 3 antioxidants-14-00665-f003:**
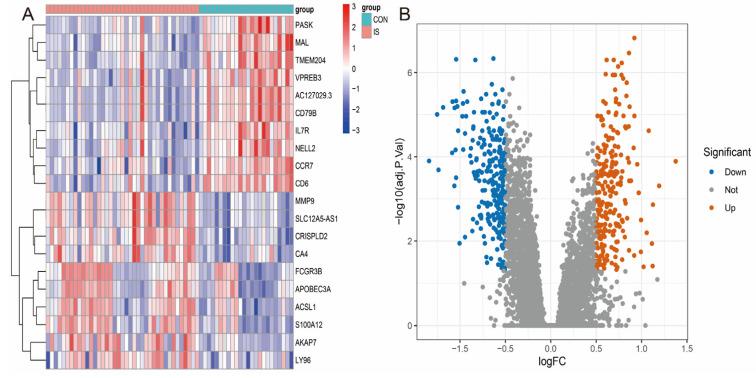
Visual analysis of differentially expressed genes. (**A**) Heatmap of differentially expressed genes. (**B**) Volcano map of differentially expressed genes.

**Figure 4 antioxidants-14-00665-f004:**
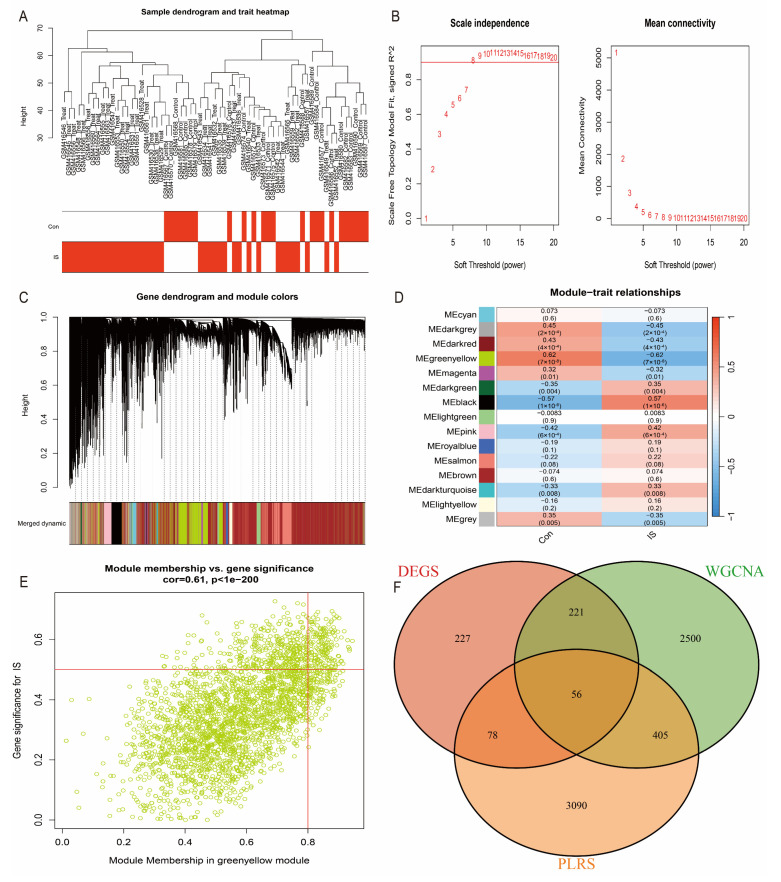
WGCNA analysis to identify key modules and hub genes. (**A**) Scale-free fit indices and average connectivity under different soft thresholds. (**B**) Gene module clustering dendrogram. (**C**) Heatmap of module-feature correlations. (**D**) Scatterplot of green-yellow modules. (**E**) Correlation of modules with clinical features. (**F**) Venn diagram of overlapping genes.

**Figure 5 antioxidants-14-00665-f005:**
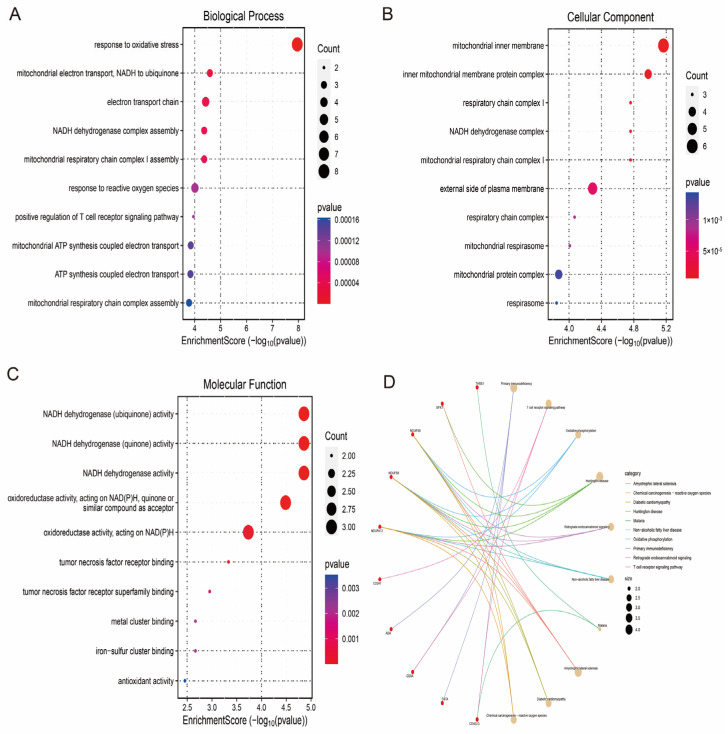
Functional enrichment analysis of the hub genes. (**A**) Biological process (BP) analysis. (**B**) Cellular component (CC) analysis. (**C**) Molecular function (MF) analysis. (**D**) KEGG pathway analysis.

**Figure 6 antioxidants-14-00665-f006:**
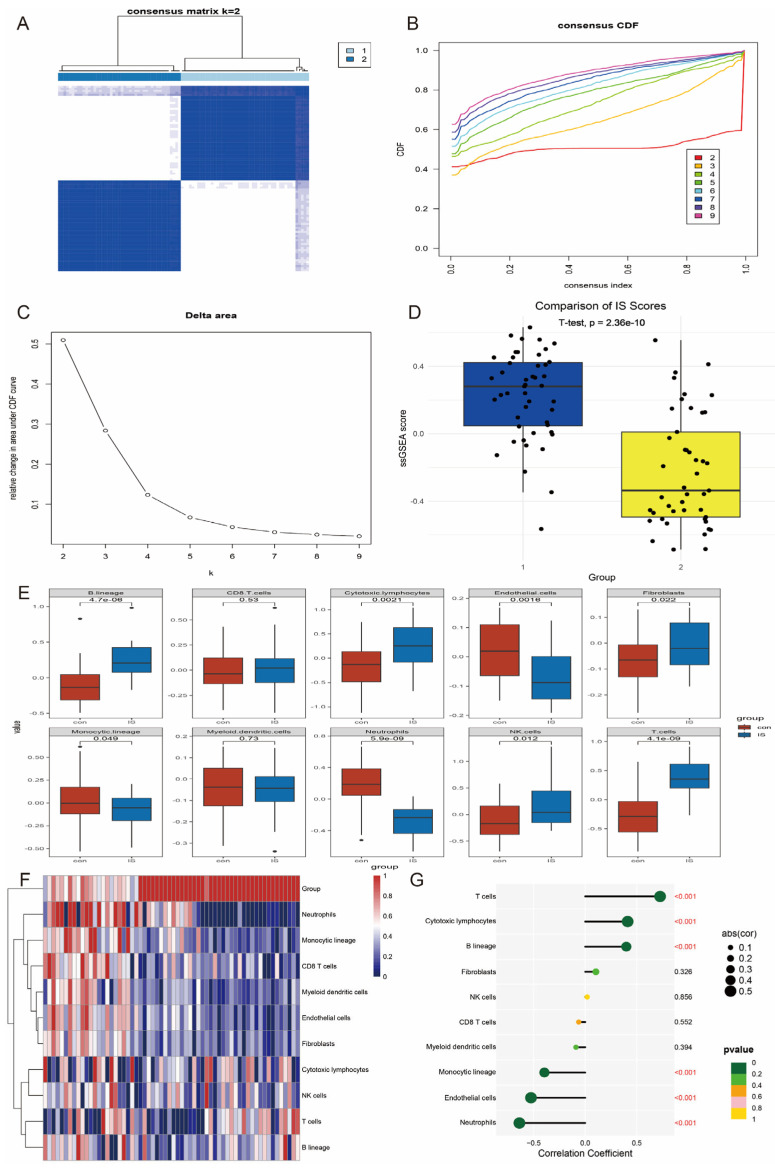
Hub gene based IS stratification and immune infiltration analysis. (**A**) Heatmap of sample clustering at k = 2 in the GSE58294 dataset. (**B**) Cumulative distribution function (CDF) curve of consensus clustering. (**C**) Relative change in area under the CDF curve. (**D**) Cluster 2 has a significantly higher IS score than cluster 1. (**E**) Immune cell correlation calculated by the MCP-counter algorithm. (**F**) Heat map of immune cell correlation. (**G**) Correlation of *GPX7* with 10 immune cells.

**Figure 7 antioxidants-14-00665-f007:**
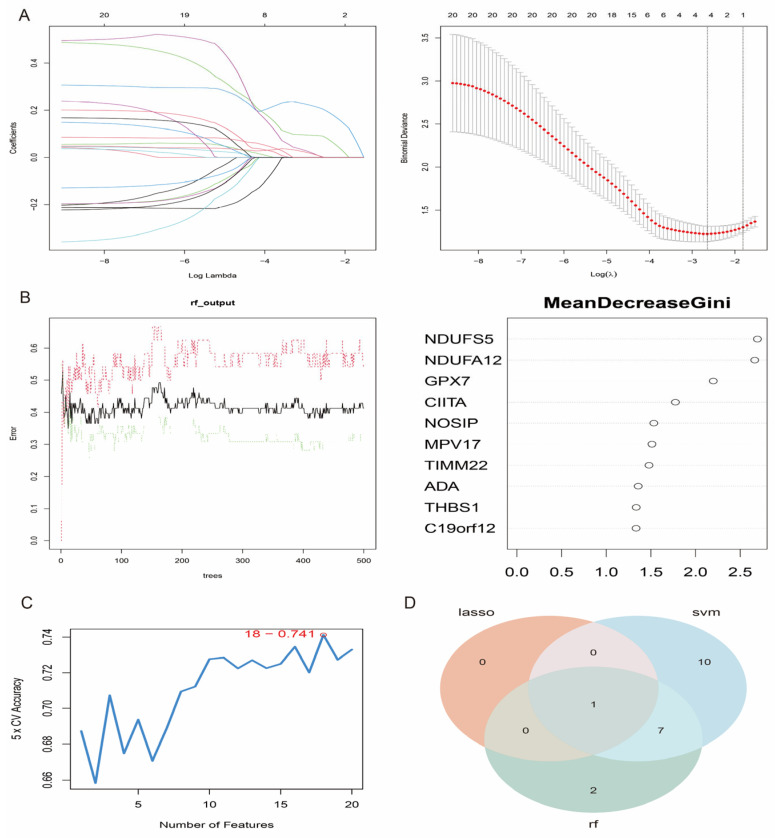
Process of selection of characteristic genes. (**A**) LASSO regression screening of trait genes. (**B**) Importance score of feature genes by random forest method. (**C**) SVM-labelled feature genes and their prediction accuracy. (**D**) Intersection analysis results of the three methods.

**Figure 8 antioxidants-14-00665-f008:**
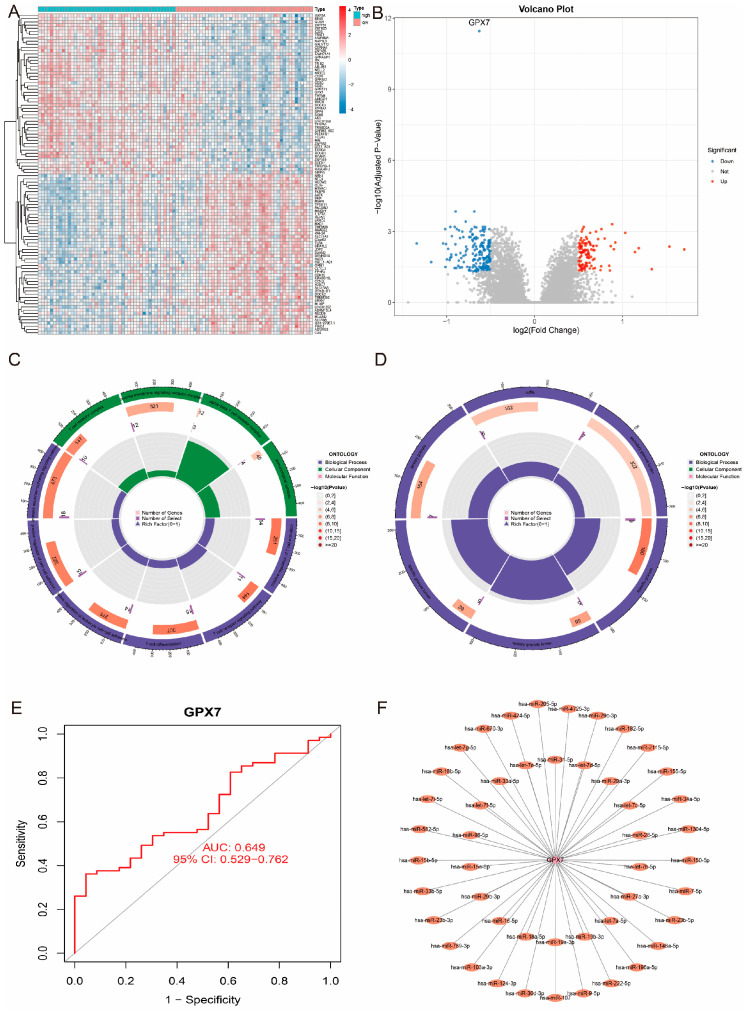
Functional analysis of *GPX7*-associated genes. (**A**,**B**) Distribution of significantly differentially expressed genes between high and low expression groups; (**C**,**D**) Results of functional enrichment analysis of high and low expression groups; (**E**) Clinical predictive ability of *GPX7* in the test set GSE58294 (AUC = 0.649); (**F**) miRNA-mRNA regulatory network.

**Figure 9 antioxidants-14-00665-f009:**
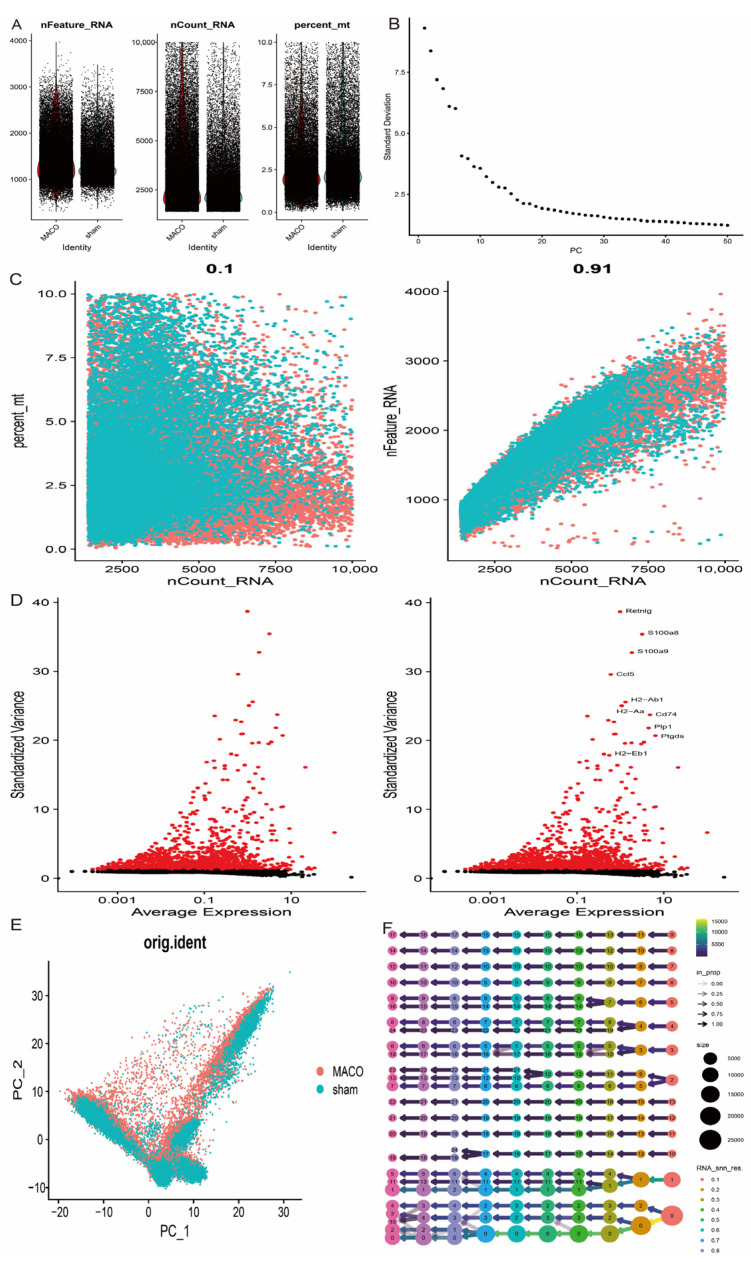
Cellular mapping of MCAO and normal tissues in GSE174574. (**A**) Key quality control metrics for single cell data; (**B**) variance ranking plot for each PC; (**C**) correlation of sequencing depth with mitochondrial content and gene number; (**D**) distribution of gene expression variability; (**E**) principal component analysis visualisation; and (**F**) selection of appropriate resolution parameter for clustering by the clustree algorithm.

**Figure 10 antioxidants-14-00665-f010:**
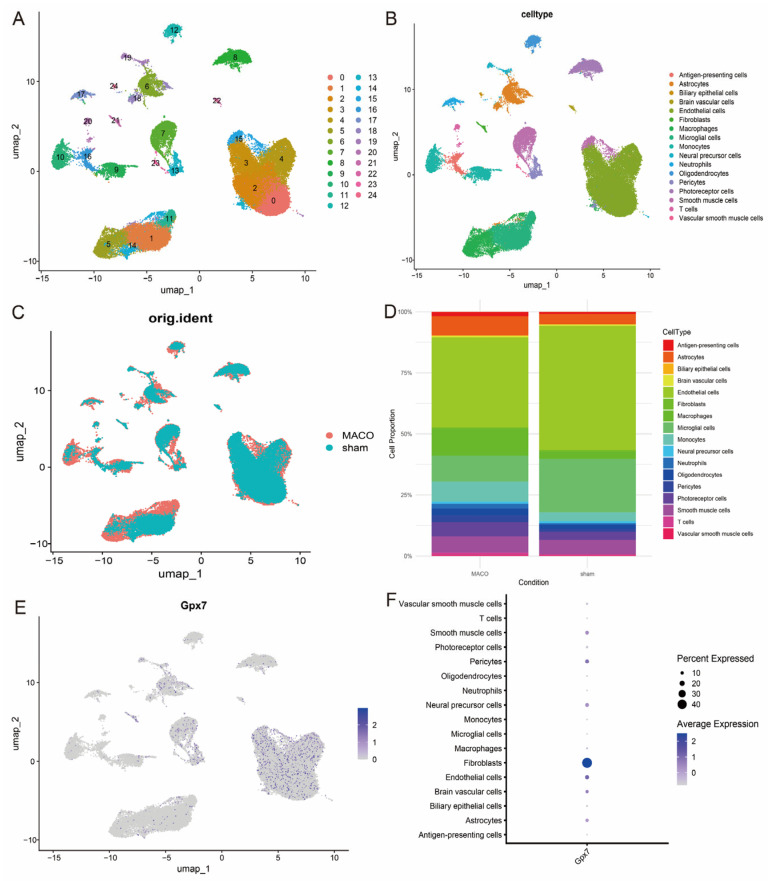
Cellular compartmentalisation and annotation. (**A**) Classification of cells into 25 clusters based on PCA and UMAP algorithms; (**B**) Annotation results of 17 cell clusters; (**C**) UMAP visualisation between the two samples; (**D**) Difference in the proportion of the 17 cell types in the two sample groups; (**E**,**F**) Expression patterns of GPX7 in different cell types.

**Figure 11 antioxidants-14-00665-f011:**
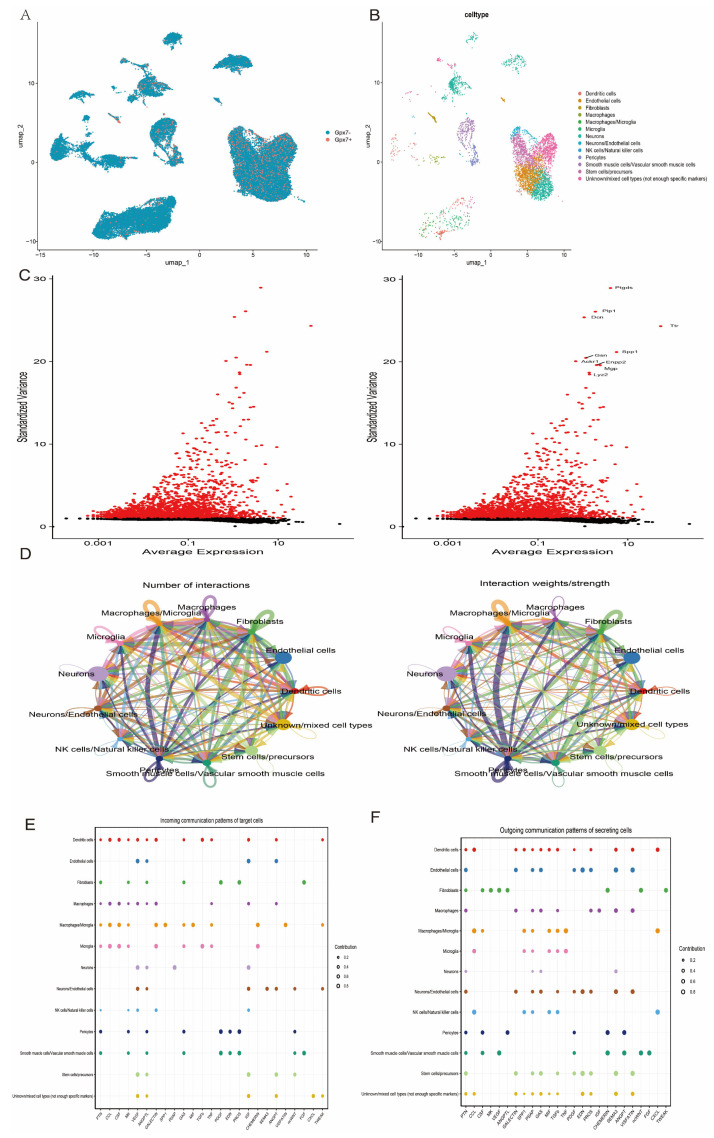
Reclustering and intercellular communication analysis of GPX7 subpopulation cells. (**A**) UMAP classification of *GPX7+* and *GPX7−* cells; (**B**) UMAP descriptions of *GPX7* subpopulation cell types; (**C**) variance plots of gene signatures with significant differences among *GPX7* subpopulation cells; (**D**) intercellular communication analysis of *GPX7* subpopulation cells; and (**E**,**F**) bubble plots illustrating intercellular communication pathways in different cells.

**Figure 12 antioxidants-14-00665-f012:**
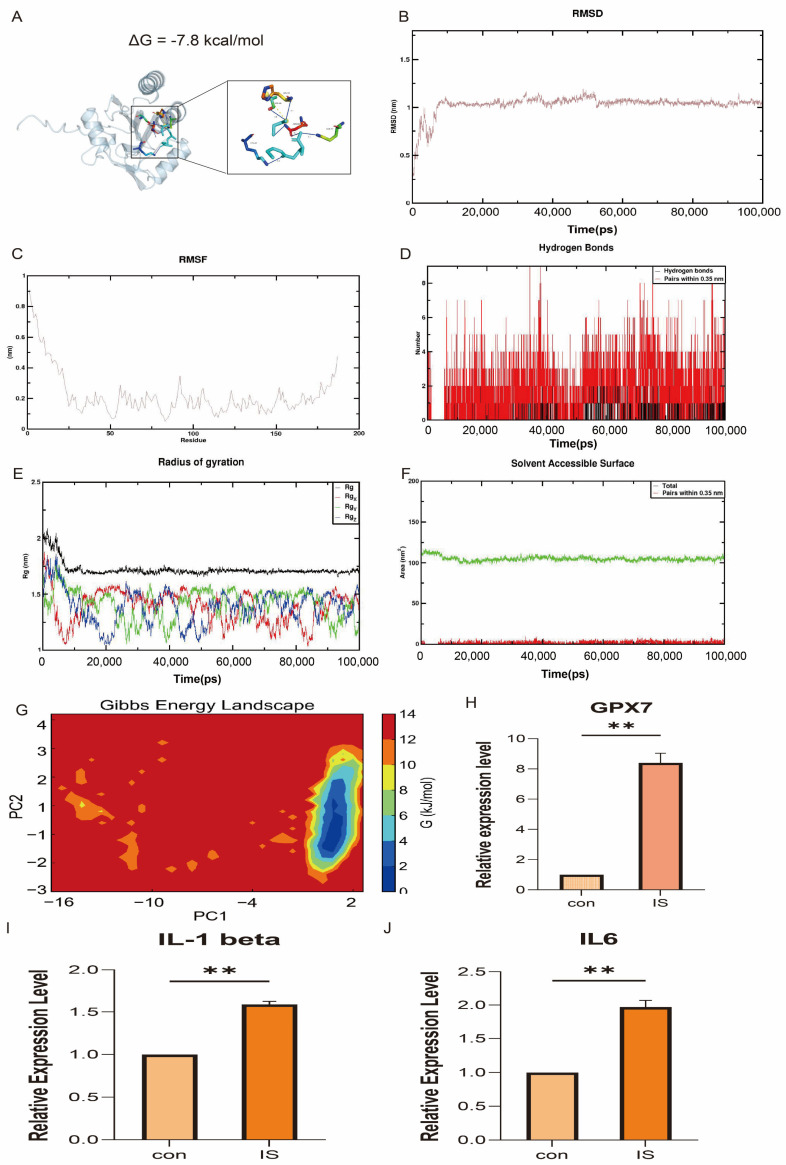
Results of molecular docking and molecular dynamics (MD) simulation analyses illustrate the stability of the three binary complexes formed. (**A**) Pymol showing the stable conformation with the lowest binding energy, (**B**) root mean square deviation (RMSD), (**C**) root mean square fluctuation (RMSF), (**D**) H-bond number, (**E**) radius of gyration, (**F**) surface area accessible to the buried solvent, and (**G**) free energy morphology maps. Expression levels of (**H**) GPX7, (**I**) IL1-beta and (**J**) IL-6 in IS and healthy controls. Data are presented as mean ± SEM. Statistical analysis was performed using two-tailed unpaired Student’s *t*-test. ** *p* < 0.01.

## Data Availability

Dataset available on request from the authors.

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
