# Peer review of "Multi-Omics and Experimental Validation Identify GPX7 and Glutathione-Associated Oxidative Stress as Potential Biomarkers in Ischemic Stroke"

_antioxidants, 2025, doi:10.3390/antiox14060665_

Round 1
Reviewer 1 Report
In this study, the analysis of the potential genes involved in ischemic stroke (IS) was performed. Many methods were used to perform the screening and the final result demonstrates the role of GPX7.
Also, in vivo experiments were executed in order to examine plasma GPX7 in the rats undergone IS.
Although the study is interesting major issues should be addressed.
Some points are reported in the above sections.
Methods: How many animals were used? What about sham operated group? There is just a mention in Results but no description in Methods.How long was the ischemic period? Why both carotid artery and middle cerebral artery were closed? Which way was euthanasia executed?
Data about histology of brain should be given. What about the extension of ischemia? Also, GPX7 and GSH should be measured in brain tissue as well.
Additional experiments should be organized in order to examine GPX7 in rats pretreated with GSH.This could add data about the potential theraputic role of them. Also, in vitro the effects of GSH on GPX7 in fibroblasts, endothelial cells, other inflammatory cells should be examined.
Only by this way the section about the "the potential of GPX7 as a therapeutic target", could be organized.
For the same reasons, the Conclusions should be changed, since there is no clear and proved evidence showing that "key role of GPX7 in the regulation of oxidative stress in ischemic stroke (IS) and its underlying molecular mechanisms". Similarly, the sentence "The results of the in vivo experiments showed that GPX7 plays an important protective role in IS models" shoudl be changed.
How many patients and controls were considered in the analysis? What about comorbidities and potential bias (sex, age, drugs, etc) which could have interferred with the obtained results?
In this study, the analysis of the potential genes involved in ischemic stroke (IS) was performed. Many methods were used to perform the screening and the final result demonstrates the role of GPX7.
Also, in vivo experiments were executed in order to examine plasma GPX7 in the rats undergone IS.
Although the study is interesting major issues should be addressed.
Some points are reported in the above sections.
Methods: How many animals were used? What about sham operated group? There is just a mention in Results but no description in Methods.How long was the ischemic period? Why both carotid artery and middle cerebral artery were closed? Which way was euthanasia executed?
Data about histology of brain should be given. What about the extension of ischemia? Also, GPX7 and GSH should be measured in brain tissue as well.
Additional experiments should be organized in order to examine GPX7 in rats pretreated with GSH.This could add data about the potential theraputic role of them. Also, in vitro the effects of GSH on GPX7 in fibroblasts, endothelial cells, other inflammatory cells should be examined.
Only by this way the section about the "the potential of GPX7 as a therapeutic target", could be organized.
For the same reasons, the Conclusions should be changed, since there is no clear and proved evidence showing that "key role of GPX7 in the regulation of oxidative stress in ischemic stroke (IS) and its underlying molecular mechanisms". Similarly, the sentence "The results of the in vivo experiments showed that GPX7 plays an important protective role in IS models" shoudl be changed.
How many patients and controls were considered in the analysis? What about comorbidities and potential bias (sex, age, drugs, etc) which could have interferred with the obtained results?
Reviewer 2 Report
The article by Tianzhi Li et al. is interesting. Some suggestions:
1. The figures contain important information, but there is too much information per figure. Can they be divided?
2. For all the information reported in the article, the discussion and conclusion are limited.
3. Since NRF2 regulates oxidative stress, could you comment in the discussion on its probable role in this reported experimental model?
4. At the end of the introduction, could you include the objective of the study?
Why does the phrase "Finally, the potential of GPX7 as a therapeutic target in IS was verified by constructing a rat middle cerebral artery occlusion (MCAO) model and combining it with qRT-PCR experiments?"
seem more like a conclusion?
The article by Tianzhi Li et al. is interesting. Some suggestions:
1. The figures contain important information, but there is too much information per figure. Can they be divided?
2. For all the information reported in the article, the discussion and conclusion are limited.
3. Since NRF2 regulates oxidative stress, could you comment in the discussion on its probable role in this reported experimental model?
4. At the end of the introduction, could you include the objective of the study?
Why does the phrase "Finally, the potential of GPX7 as a therapeutic target in IS was verified by constructing a rat middle cerebral artery occlusion (MCAO) model and combining it with qRT-PCR experiments?"
seem more like a conclusion?
Round 2
Reviewer 1 Report
The authors have addressed many of the raised points. However, some issues should be still faced.
How many patients and controls were considered in the analysis? What
about comorbidities and potential bias (sex, age, drugs, etc) which could have
interferred with the obtained results?
The first part of the paper is about "peripheral blood RNA sequencing data of IS patients were obtained from the GEO database (https://www.ncbi.nlm.nih.gov/geo/) for systematic analysis of 80 the function of oxidative stress-related genes and their role in ischemic stroke (IS)"
The results show that "To develop a gene signature-based stratification method for ischemic stroke (IS), we extracted 1719 oxidative stress-related genes from the GeneCards database and calculated the expression profiles of these genes in the GSE58294 dataset by scoring them using the ssGSEA (single-sample gene set enrichment analysis) method. As shown in Figure 2A, ssGSEA analysis revealed the expression patterns of these oxidative stress-related genes. Further statistical analyses showed that the oxidative stress scores in the patient group were significantly higher than those in the control group (P = 0.047; Figure 2B). This result suggests that oxidative stress-related scores have the potential to discriminate between IS patients and healthy individuals".
In order to get more precise results about this issue, some clinical data should be also considered in the analysis about the involvement of GPX7. They could represent bias.
Also some experimental evidences showing the existence of ischemic brain damages should be given.
In Statistical analysis " One-way analysis of variance (ANOVA) was used for comparisons between more than two groups"...when was ANOVA used? Details about the use of statistical test should be added in figure legends.
The authors have addressed many of the raised points. However, some issues should be still faced.
How many patients and controls were considered in the analysis? What about comorbidities and potential bias (sex, age, drugs, etc) which could have interferred with the obtained results?
The first part of the paper is about "peripheral blood RNA sequencing data of IS patients were obtained from the GEO database (https://www.ncbi.nlm.nih.gov/geo/) for systematic analysis of 80 the function of oxidative stress-related genes and their role in ischemic stroke (IS)"
The results show that "To develop a gene signature-based stratification method for ischemic stroke (IS), we extracted 1719 oxidative stress-related genes from the GeneCards database and calculated the expression profiles of these genes in the GSE58294 dataset by scoring them using the ssGSEA (single-sample gene set enrichment analysis) method. As shown in Figure 2A, ssGSEA analysis revealed the expression patterns of these oxidative stress-related genes. Further statistical analyses showed that the oxidative stress scores in the patient group were significantly higher than those in the control group (P = 0.047; Figure 2B). This result suggests that oxidative stress-related scores have the potential to discriminate between IS patients and healthy individuals".
In order to get more precise results about this issue, some clinical data should be also considered in the analysis about the involvement of GPX7. They could represent bias.
Also some experimental evidences showing the existence of ischemic brain damages should be given.
In Statistical analysis " One-way analysis of variance (ANOVA) was used for comparisons between more than two groups"...when was ANOVA used? Details about the use of statistical test should be added in figure legends.
